# Inspired by Earthworms and Leeches: The Effects of Cylindrical Pit Arrays on the Performance of Piston-Cylinder Liner Friction Pairs

Tianyu Gao [1] , Hao Chen [1], Danna Tang [2] and Yumo Wang [1,*]

[1] School of Intelligent Manufacturing, Nanjing University of Science and Technology, Nanjing 210094, China; gty@njust.edu.cn (T.G.); chenhaoyl@163.com (H.C.)

[2] School of Mechanical Engineering, Changshu Institude of Technology, Suzhou 215506, China; dana.cug@foxmail.com

[*] Correspondence: wangyumo@njust.edu.cn

**Abstract:** To improve the friction and wear performance of the piston-cylinder liner friction pair, inspired by earthworms and leeches, 27 kinds of pistons with cylindrical pit arrays are designed and processed. Through a friction test, four superior textured pistons are optimized, and wear, life and thermal imaging tests are performed. Finite element analysis of the friction pair model is performed, and the friction and wear mechanisms are discussed. The results show that the pistons with cylindrical pit arrays have excellent friction and wear performance, less heat generation by friction, longer lives and less scratches on the cylinder liner. The temperature of the optimized textured pistons was reduced by around 5–10 °C. The wear amount of some textured pistons was reduced by over 50%, resulting in an improvement in their lifespan of at least 30% or more. The results of the finite element analysis indicate that the textured piston exhibited reduced deformation and favorable stress–strain distribution and satisfied the required contact pressure.

**Keywords:** bionic inspiration; friction; wear; mud pump; piston; temperature

## 1. Introduction

Reciprocating mud pumps, due to their strong versatility, have a wide range of applications in agricultural production, construction engineering and energy drilling. In the field of agriculture, mud pumps play an integral role in land reclamation, which is commonly seen in coal mine areas. Reclamation through mud pumps involves using water pumps to generate high-pressure water jets to wash dry soil. The broken soil forms a slurry mixture with water, which is then transported to the land reclamation area through mud pumps and pipelines. Hydraulic dredging units are vital hydraulic machinery in the fish farming industry, consisting of mud pumps, high-pressure water guns and delivery pipes used for digging or deepening fish ponds. In the field of construction, mud pumps can not only carry out reservoir silt removal and silt-solidified river dike construction through water flushing excavation but also transport clear water and cement in foundation construction. In the field of energy drilling, mud pumps are crucial in geological exploration and oil drilling. Mud pumps serve as the core of the drilling fluid circulation system, providing high-pressure liquid to flush the drill bit and discharge the drilling fluid and thus greatly improving drilling efficiency.

As the main component and major wearing parts of the mud pump equipment, the piston-cylinder liner friction pair has an important meaning to the stability and safety of the mud pump. The friction and wear performance and the temperature of the friction pair are important indicators of the mud pump which determine the life of the piston and the cylinder liner. How to reduce the friction and wear of the friction pair has always been a concern. The mud pump piston is made of rubber, and the cylinder liner is made of steel.

Rubber materials are widely used in various friction and sealing parts. Relevant scholars' research on the friction and wear of these parts has mainly focused on rubber seals [1–4], rubber tires [5,6] and rubber conveyor belts [7]. For instance, Zhang et al. conducted research on O-rings [1]. They studied the effects of the pre-compression amount, fluid pressure and friction coefficient on the static and dynamic sealing performance of O-rings. Moreover, Zhou et al. delved into the significant impact of pressure on the rubber-sealing capabilities within an abrasive drilling environment [4]. They meticulously analyzed the wear process and mechanisms resulting from the invasion of micro-rock debris at the sealing interface. Additionally, they compared the characteristics of particle disintegration and the evolution of wear under varying conditions of high and low pressure. There are also unconventional rubber products, such as how Ke et al. studied both the kinetic friction characterizations and the stick–slip motion phenomena for tubular rubber seals [8]. There are few pieces of research on the friction pairs of pumps, and they are mainly focused on friction pairs in the plunger pump [9–12]. Moreover, Jia et al. designed a composite texture combining micro-concave pits with variable depth grooves to improve the sealing performance of diesel engine piston pairs and reduce wear [13]. The research results show that the composite texture can produce an appropriate oil film thickness to resolve insufficient lubrication issues. The composite texture improved the lubrication and sealing performance of the piston, reducing the wear depth and average roughness. Bench tests also proved an enhancement in the sealing effect. Plunger pump and engine pistons [14], including piston rings as friction pairs [15], are all made of metal materials [16,17]. There is limited research on non-metallic material pistons such as rubber and polyurethane pistons. Research on rubber pistons mainly focuses on medical injection devices or other scenarios that implement seals with rubber materials. Studies on mud pump pistons are scarce.

With the development of science and technology, it is important to extend the service life by reducing the friction and wear of the wearing parts of the friction pair. The texture treatment of the surface of the friction pair has been extensively studied and proven to be an effective method to improve the tribological properties of the interface. The surface texture can reduce friction, improve lubrication conditions and reduce wear. Etsion found a variety of surface textures that can significantly improve the performance of friction pairs [18,19]. These surface structures mainly include pit structures [20,21], groove structures [22–26] and other structures [27–30]. The pit structure includes circular pits, oval pits and square pits [31–35].

This paper designs and processes a mud pump piston with an array of cylindrical pits inspired by earthworms and leeches. A friction test bed for a piston-cylinder liner friction pair was built, and the friction was tested. A better piston is optimized for wear and life testing. Then, the temperature of the piston-cylinder friction pair is tested and analyzed. Finally, a finite element analysis model is established to explain the friction and wear mechanism, and the wear of the piston and the scratches on the cylinder liner are discussed. This research can provide a theoretical basis for the study of friction pairs of mud pumps. The research results are of great significance to the design and evaluation of mud pump pistons. This paper also offers valuable insights into the surface texture design of materials like rubber and polyurethane. The optimized texture parameters, coupled with finite element analysis, provide robust data and theoretical underpinnings for further investigation in this field.

## 2. Experimental

### 2.1. Piston Design and Test Scheme

Nature has provided a lot of inspiration for the development of human science. Organisms have evolved many marvelous structures to adapt to their environments. For example, as shown in Figure 1, earthworms secrete mucus through holes on their backs to reduce friction when entering the soil. Similarly, leeches secrete mucus from their surfaces to facilitate smooth entry into mud, as depicted in Figure 2. Furthermore, regularly spaced array structures have been proven to reduce friction, but they are less commonly applied in

elastic rubber materials. Mud pump pistons are typically made of softer elastic materials such as rubber and polyurethane, and they transport media like mud or clean water, which are highly similar to the living environments of earthworms and leeches. The design of the cylindrical pit array on the piston surface in this article is inspired by earthworms and leeches. The surface back holes of earthworms are typically arranged in a single row, while leeches exhibit a distribution pattern of both single rows and staggered arrays on their surfaces. Taking inspiration from these distribution patterns, we have designed a textured piston.

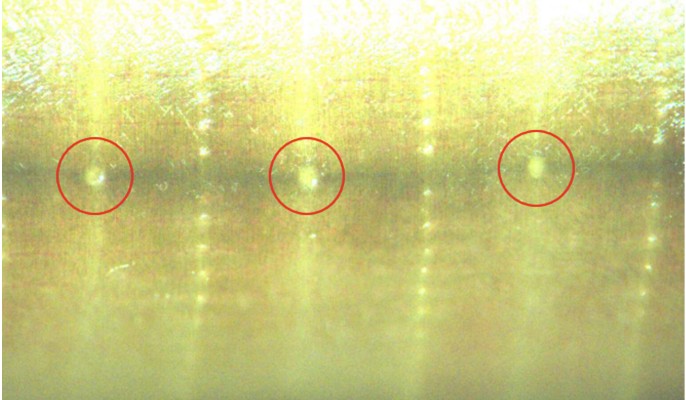

**Figure 1.** Back holes of earthworms.

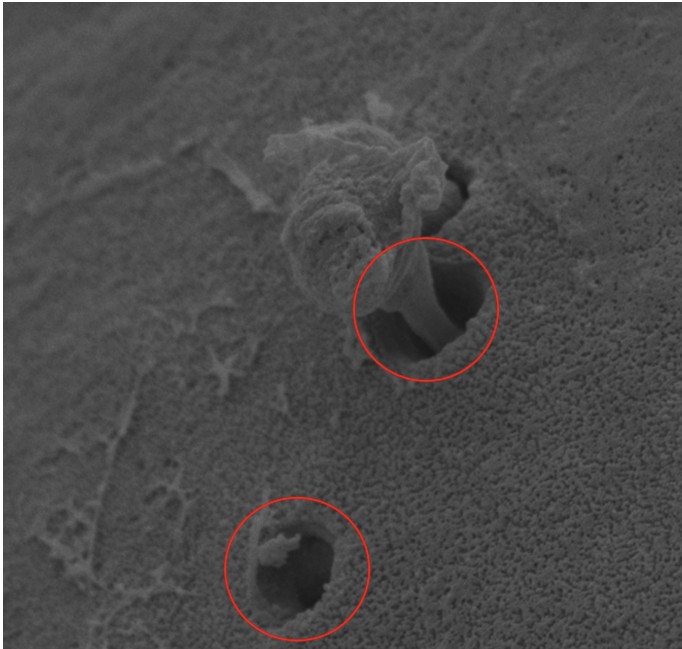

**Figure 2.** SEM images of a leech's surface.

The BW-250 mud pump piston mainly consists of a nylon part and a rubber part. The cylindrical pit array design on the surface of the rubber part of the piston is shown in Figure 3a. The cylindrical pit includes two parameters: the cylindrical height $H$ and the cylindrical diameter $D$ (Figure 3b). The distribution and density of the cylindrical pit array have a significant effect on the performance of the piston. The array of cylindrical pits in this test is designed as three rows with a staggered distribution, and the number of cylindrical pits in each row is $N$. The test factors and their levels are shown in Table 1. In this study, we utilized an experimental optimization design methodology, which is extensively employed in various research fields. As an example, Miladinović et al. applied the L18 design involving 18 experimental runs [36]. In comparison, our study adopted the

L27 design, which involves 27 experimental runs. This approach allowed us to gather a more comprehensive dataset for analysis and draw robust conclusions.

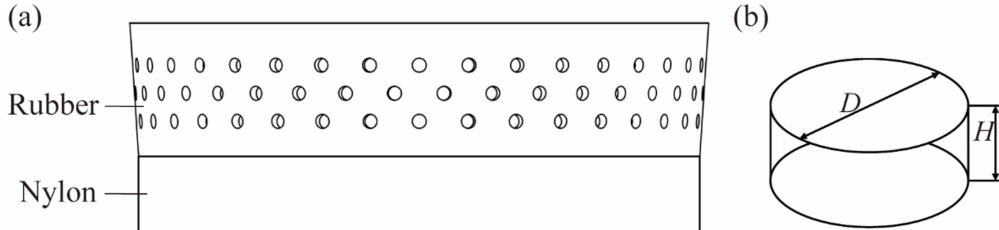

**Figure 3.** Piston design: (**a**) cylindrical pit array and (**b**) cylindrical pit parameters.

**Table 1.** Test factors and their levels.

| Levels | Factors | | |
|---|---|---|---|
| | $D$ (mm) | $H$ (mm) | $N$ |
| 1 | 1 | 0.5 | 36 |
| 2 | 1.5 | 1.0 | 24 |
| 3 | 2 | 1.5 | 18 |

In this article, we utilize a computerized numerical control (CNC) machining center to individually fabricate cylindrical pit textures on the rubber surface of the standard BW-250 mud pump piston, as illustrated in Figure 4a. Processing textures with varying diameters $D$ requires the replacement of micro-drill bits specific to each diameter. The processed textured pistons from 27 groups are labeled according to the test number $X$ in Table 2, as shown in Figure 4b.

**Table 2.** Friction test results.

| $X$ | $H$ (mm) | $D$ (mm) | $N$ | $F+$ (N) | $F-$ (N) | $X$ | $H$ (mm) | $D$ (mm) | $N$ | $F+$ (N) | $F-$ (N) |
|---|---|---|---|---|---|---|---|---|---|---|---|
| 1 | 0.5 | 1.0 | 36 | 121.84 | 152.12 | 15 | 1.0 | 1.5 | 18 | 116.02 | 158.70 |
| 2 | 0.5 | 1.0 | 24 | 158.03 | 184.00 | 16 | 1.0 | 2.0 | 36 | 128.95 | 152.75 |
| 3 | 0.5 | 1.0 | 18 | 141.66 | 212.47 | 17 | 1.0 | 2.0 | 24 | 157.37 | 193.41 |
| 4 | 0.5 | 1.5 | 36 | 105.46 | 119.84 | 18 | 1.0 | 2.0 | 18 | 144.72 | 201.28 |
| 5 | 0.5 | 1.5 | 24 | 134.96 | 187.88 | 19 | 1.5 | 1.0 | 36 | 142.32 | 167.14 |
| 6 | 0.5 | 1.5 | 18 | 119.53 | 185.47 | 20 | 1.5 | 1.0 | 24 | 154.65 | 188.10 |
| 7 | 0.5 | 2.0 | 36 | 146.78 | 162.46 | 21 | 1.5 | 1.0 | 18 | 174.18 | 197.67 |
| 8 | 0.5 | 2.0 | 24 | 122.94 | 174.28 | 22 | 1.5 | 1.5 | 36 | 109.86 | 119.95 |
| 9 | 0.5 | 2.0 | 18 | 145.29 | 215.25 | 23 | 1.5 | 1.5 | 24 | 161.89 | 184.65 |
| 10 | 1.0 | 1.0 | 36 | 111.34 | 137.58 | 24 | 1.5 | 1.5 | 18 | 171.79 | 213.73 |
| 11 | 1.0 | 1.0 | 24 | 175.99 | 215.31 | 25 | 1.5 | 2.0 | 36 | 116.40 | 130.25 |
| 12 | 1.0 | 1.0 | 18 | 151.56 | 215.28 | 26 | 1.5 | 2.0 | 24 | 162.17 | 202.77 |
| 13 | 1.0 | 1.5 | 36 | 111.02 | 117.76 | 27 | 1.5 | 2.0 | 18 | 149.03 | 192.84 |
| 14 | 1.0 | 1.5 | 24 | 155.62 | 174.00 | S | 0 | 0 | 0 | 196.22 | 251.16 |

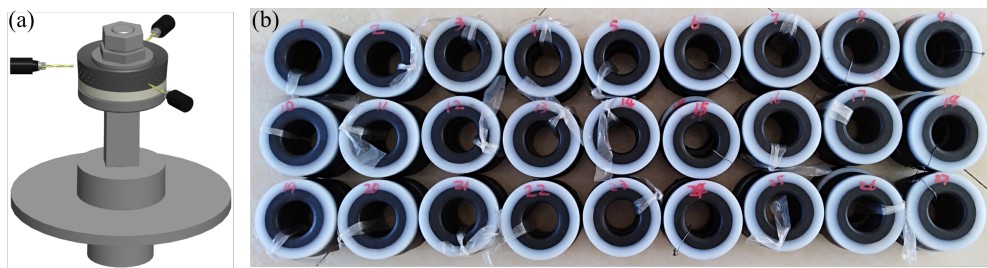

**Figure 4.** The (**a**) fabrication and (**b**) labeling of textured pistons.

### 2.2. Test Bench and Test Method

Firstly, the friction force of the standard piston and the piston with a cylindrical pit array was measured by a friction force test bench (Figure 5) built in the laboratory. For the test method, we first applied grease evenly to the contact surface between the piston and the cylinder liner to facilitate adequate lubrication within the contact surfaces of the friction pair and the cylindrical pits of the textured piston. Then, the test piston was assembled on the pull rod. On the other side, a guiding piston was paired to create a sealed chamber between the two pistons. Subsequently, the sealed chamber was filled with mud, and the mud composition followed the relevant provisions of the national industry standard "Reciprocating Mud Pump for Geological Drilling" (DZ/T 0090-2017). The experimental mud possessed a density of 1.306 g/cm$^3$ and a sand content of 2.13%. The experimental pressure was controlled at 0.1 MPa, achieved by employing a pressurization device in conjunction with a pressure gauge. When the driving device was turned on, the pull rod drove the piston to reciprocate. The motor speed was set to 2.67 r/min. The experimental methods and conditions mentioned above were essentially consistent with the actual operating conditions of the BW-250 mud pump. The friction force of the test piston was measured by a pull-press sensor. The friction force experiments were conducted indoors at a temperature of approximately 20 °C. To ensure the accuracy of the experimental data, after each test, it was necessary to wait for the test bench and mud to cool down to room temperature before proceeding with the next test. Friction force tests were conducted for both the standard piston and the 27 groups of textured pistons. The friction force data for the tested pistons were collected after the test equipment had been running steadily for 30 min. Finally, the friction force data were exported using an AD conversion device, and the average friction force during the reciprocating stroke was calculated.

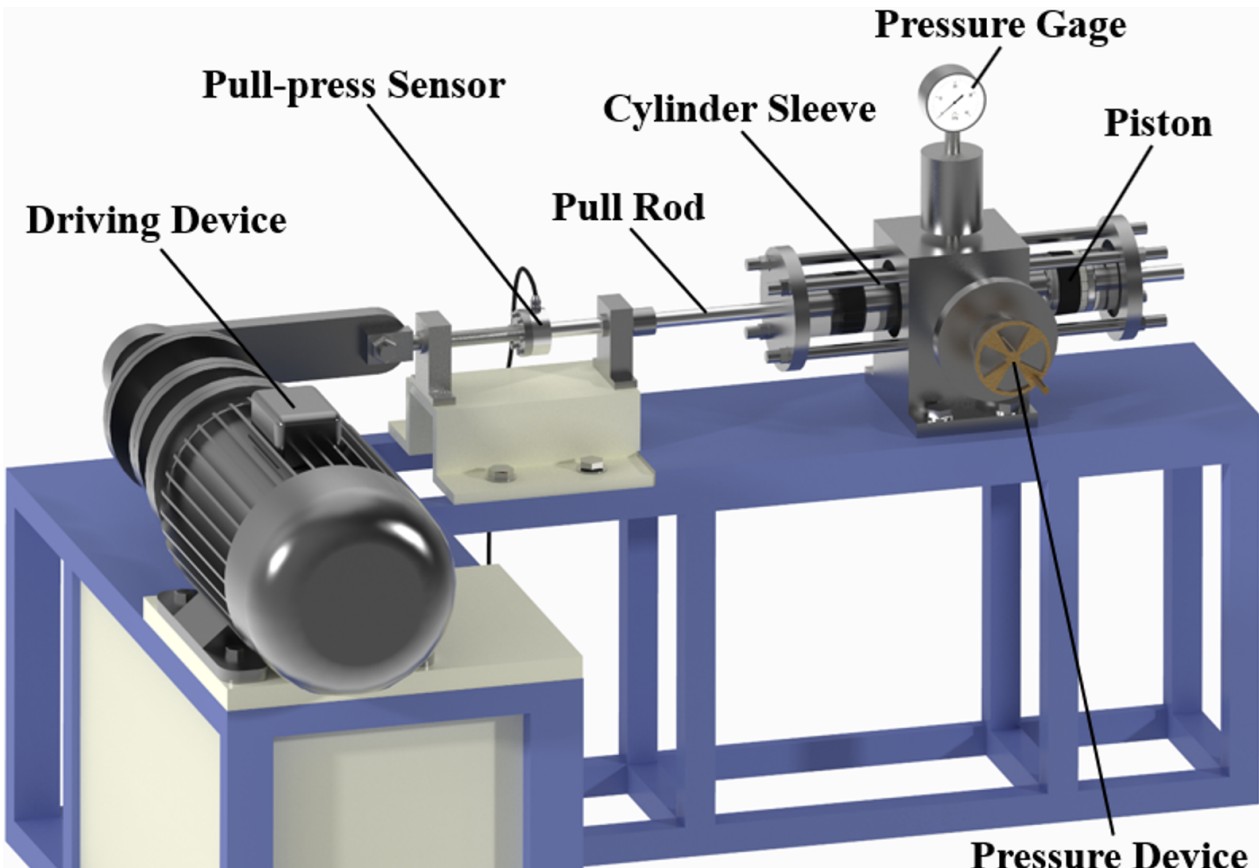

**Figure 5.** Friction force test bench.

According to the results of the friction test, the parameters of the optimized array of cylindrical pits were obtained. The life test, wear test and thermal imaging test of the optimized piston were carried out on the BW-250 mud pump test equipment (Figure 6). The BW-250 mud pump test equipment includes a drive motor, gearbox, pump body, pressure valve, pressure gauge, inlet pipe and outlet pipe. Our research object was the piston and cylinder liner in the pump body, as shown in Figure 7. Consistent with the friction test bench, the pull rod drove the piston to reciprocate in the cylinder liner under the drive of the motor.

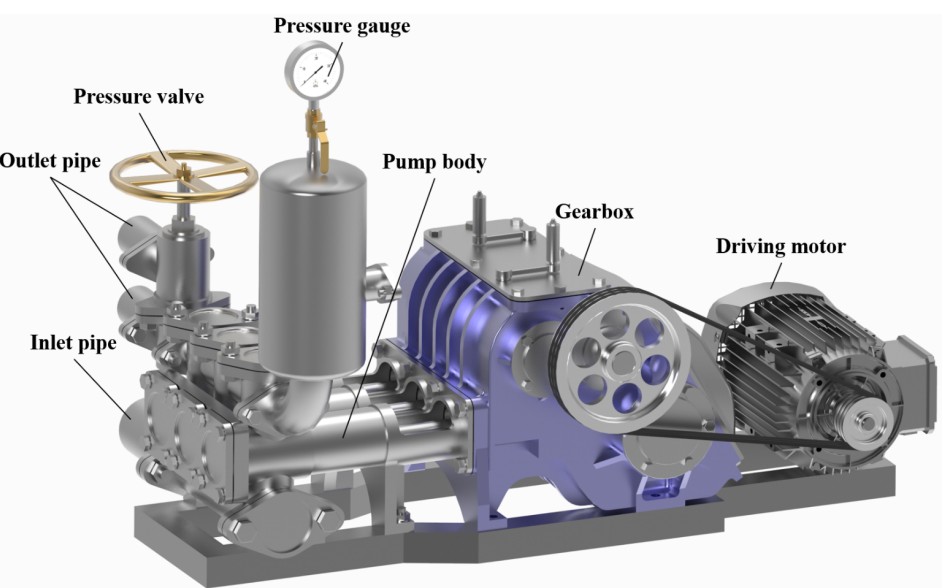

**Figure 6.** BW-250 mud pump test equipment.

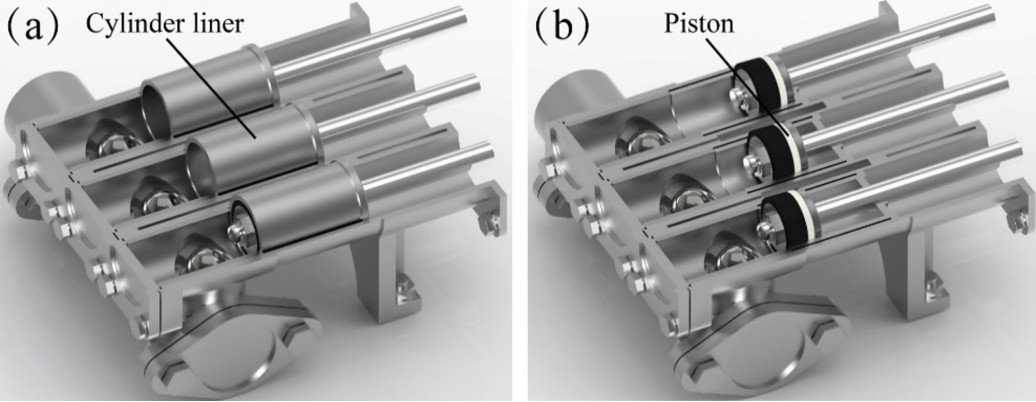

**Figure 7.** Pump body: (**a**) cylinder liner and (**b**) piston.

As the BW-250 mud pump is a three-cylinder pump, it can test three pistons at the same time. The test was divided into two groups, and each group was equipped with a standard piston.Pistons 4, 13 and S-1 # (standard piston) were in the first group. Pistons 22, 25 and S-2 # (standard piston) were in the second group. Wear and life tests were carried out in each group. The mud pump test gear was set to A2-B3. The test stroke was 200 strokes/min, the displacement was 250 L/min, and the pressure was 0.5 MPa.

We calculated the wear rate ($w$) of the piston using the formula

$$w = \frac{m_0 - m_f}{m_0} \times 100\% \tag{1}$$

where $m_0$ is the mass before wear and $m_f$ is the mass after wear. Calculate the percentage ($v$) increase in the life of the piston using the formula

$$v = \frac{L_1 - L_0}{L_0} \times 100\% \tag{2}$$

where $L_0$ is the life of the standard piston and $L_1$ is the life of the piston with cylindrical pit arrays.

Temperature is an important factor affecting the friction and wear of rubber. Friction heat generation between the piston and cylinder liner has an important influence on piston wear. A Fluke Ti400+ thermal imaging camera was used to analyze the temperature variations in the friction pair, as depicted in Figure 8a. In this study, the average temperature of a specific region was obtained by capturing thermal images and selecting corresponding measurement areas in three different cylinder liners, as shown in Figure 8b.

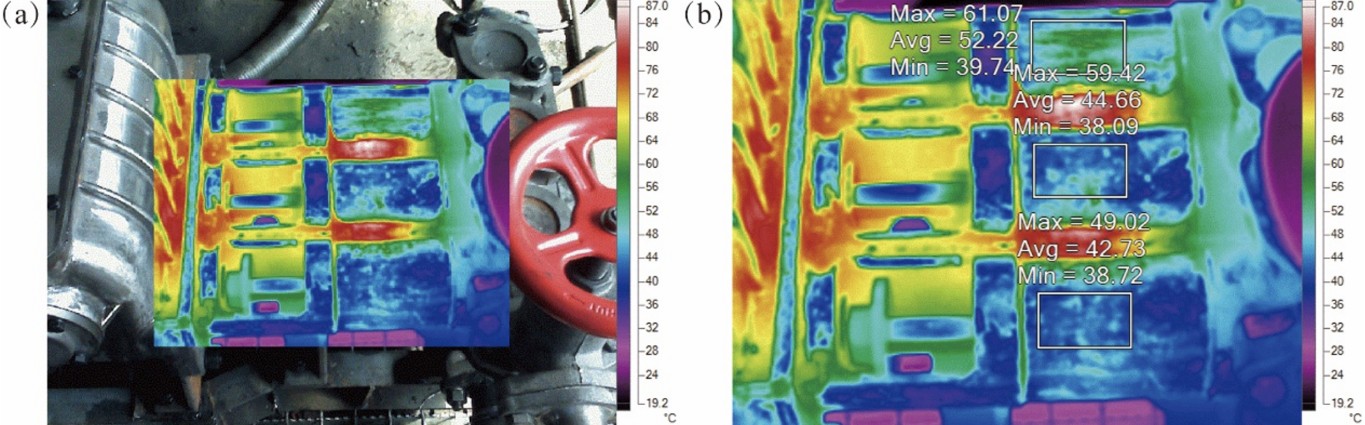

**Figure 8.** Thermal imaging chart: (**a**) friction pair and (**b**) temperature measurement area.

The friction and wear mechanism of a mud pump piston is a complicated process which includes many factors such as deformation, stress and strain. In order to explore the mechanism, we model the standard piston and textured pistons to carry out finite element analysis. Using Cero 9.0 software, a modeling process was conducted to create 27 different textured pistons based on the structural dimensions of the standard piston and cylindrical pit array design, as shown in Figure 9.

The models of the standard piston and textured piston were saved as xt files in Cero and imported into the Geometry module of the ANSYS Workbench. Subsequently, the friction pair model of the piston-cylinder liner was simplified, and the components of the friction pair, including the piston, cylinder liner and steel core, were named accordingly in the Design Modeler module. Considering that the piston-cylinder friction pair constitutes a symmetrical model, we utilized the Design Modeler module to remove three-fourths of the piston-cylinder friction pair assembly to reduce the computational time required for finite element analysis. The leftover one-fourth of the model was then subjected to simulation and calculation, as shown in Figure 10.

The finite element analysis method in the ANSYS Workbench comprises three main steps: preprocessing, solving and post-processing. In the preprocessing stage, it is essential to determine the geometric region and physical properties of the actual problem or system to be solved. Subsequently, various parameters for the elements must be defined, such as the element type, material attributes, geometric properties (shape and size), connectivity and basis functions. Lastly, the boundary conditions and applied loads for the system need to be specified. The following will detail the boundary conditions set for the finite element analysis in this study. Firstly, we navigated to the Analysis Systems directory and selected the Static Structural module to perform the finite element analysis. Subsequently, the material properties of the piston-cylinder liner friction pair were set. In the Engineering

Data module, the rubber material parameters were defined for the rubber component using a two-parameter Mooney–Rivlin model. The material constant C10 parameter was set to 2.5 MPa, and the material constant C01 parameter was set to 0.625 MPa. The material properties of the cylinder liner and steel core, excluding the piston, were set to the default structure steel. Moving forward, the contact types for the piston-cylinder liner friction pair were configured. The contact type between the piston and steel core was set to "No Separation", whereas the contact type between the piston and cylinder liner was defined as "Frictional". Asymmetric behavior was selected for both contacts, utilizing the "Augmented Lagrange" formulation. Subsequently, the mesh for the friction pair between the piston and cylinder liner was generated. The automatic meshing method was employed, and specific attention was given to refining the mesh for the piston contact surface and the cylindrical pit arrays, as shown in Figure 11.

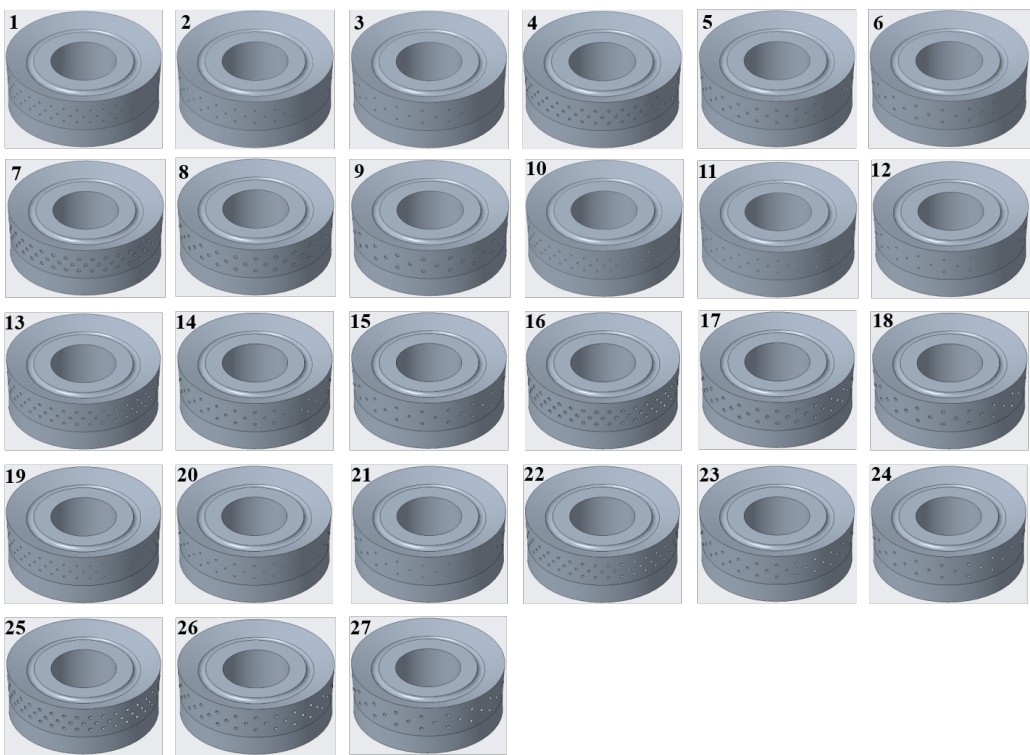

**Figure 9.** Modeling of textured pistons.

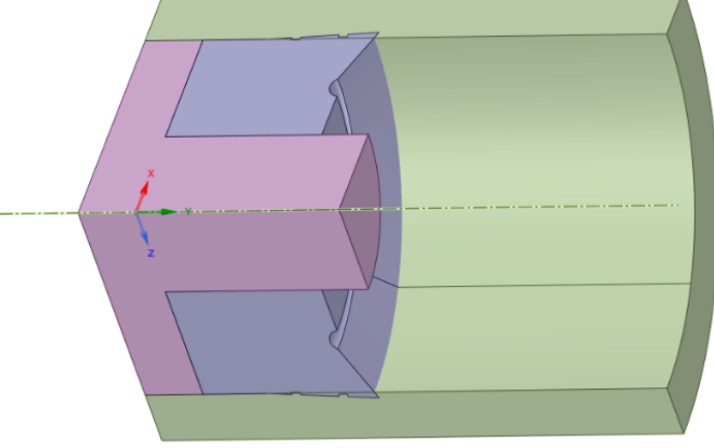

**Figure 10.** Quarter model.

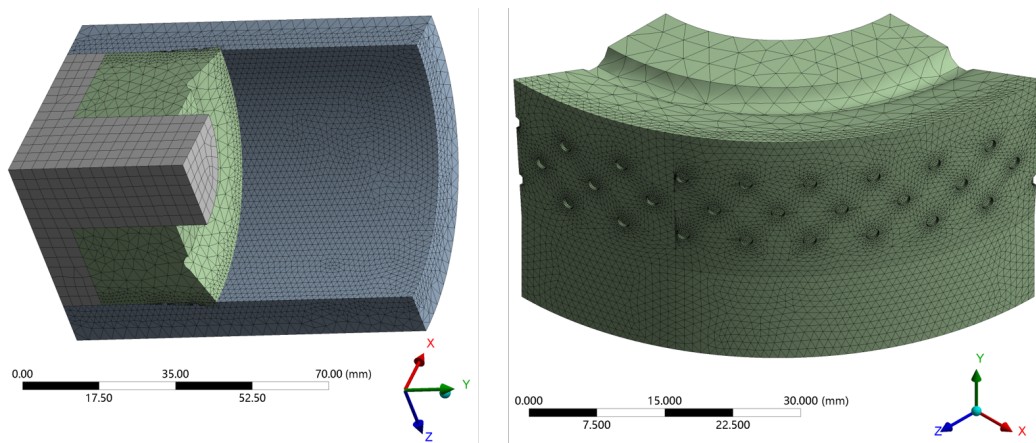

**Figure 11.** Mesh refinement.

Subsequently, boundary constraints were imposed on the piston-cylinder friction pair as follows. Fixed constraints were placed at both ends of the cylinder liner, symmetric and frictionless support constraints were established on the cut surface, pressure constraints were applied to the piston's pressure-bearing surface, and displacement conditions were set for the load end of the steel core, as depicted in Figure 12. Finally, the desired solutions for the deformation, stress–strain and contact pressure were defined. Once these steps are completed, one can click on "solve" to execute the computation and review the solution results through the post-processing module after the computation has finished.

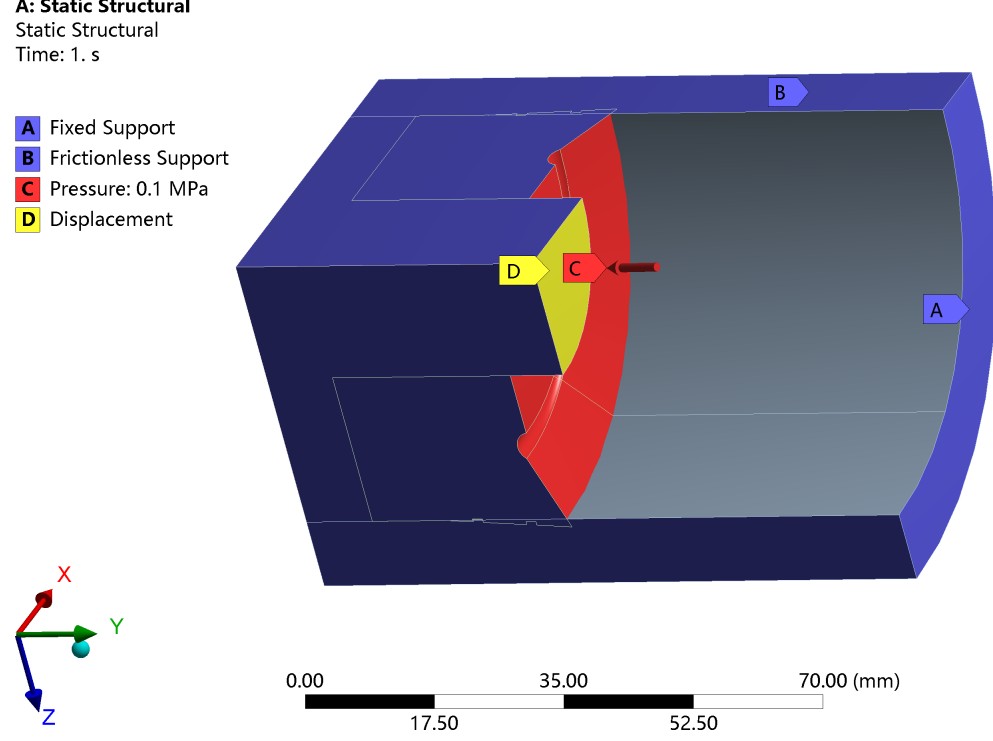

**Figure 12.** Boundary conditions.

## 3. Results and Discussion

### 3.1. Friction Test Results

The friction test results are shown in Table 2. *X* is the test number, where 1–27 is the test number of the piston with a cylindrical pit array, and *S* is the test number of the standard piston. *F*+ is the average friction force in the forward stroke of the piston, and *F*− is the average friction force in the reverse stroke of the piston. It can be seen that, compared

with the standard piston, the $F+$ and $F-$ values of pistons 1–27 were both reduced to some extent. Among them, $F+$ and $F-$ for tests 4, 13, 22 and 25 reduced the most. Their $F+$ drag reduction rates were 46.25%, 43.42%, 44.01% and 40.68%, respectively. Their $F-$ drag reduction rates were 52.29%, 53.11%, 52.24% and 48.14%, respectively.

When $D$, $H$ and $N$ are unchanged, $F+$ and $F-$ are shown in Table 3. It can be seen that as $D$ increased, both $F+$ and $F-$ decreased first and then increased, and a minimum was achieved at $D = 1.5$ mm. Therefore, in order to improve the friction performance of the piston, $D$ should not be too large or too small. $F+$ increased with the increase in $H$, while $H$ had no significant effect on $F-$. In Table 3, an increase in $H$ leads to a notable variation in the average value of $F+$, ranging from 7 N to 10 N. However, the average value of $F-$ exhibits minimal changes, remaining distributed within the range from 174 N to 177 N, with a difference of approximately 3 N. Therefore, an excessive $H$ value has adverse effects on the friction performance of the piston. When $H$ reached its maximum limit of 1.5 mm, both $F+$ and $F-$ exhibited their highest average values, which is not ideal for our purposes. Taking into account the overall trend in the variations of $F+$ and $F-$, a value of $H$ equal to 0.5 mm was deemed most advantageous for optimizing the piston's friction performance. $F+$ and $F-$ both achieved the minimum value at $N = 36$, which means that the friction performance of the piston was better when $N$ was larger.

**Table 3.** $F+$ and $F-$ when $D$, $H$ and $N$ are unchanged.

|  | $D$ (mm) | | | $H$ (mm) | | | $N$ | | |
|---|---|---|---|---|---|---|---|---|---|
|  | 1.0 | 1.5 | 2.0 | 0.5 | 1.0 | 1.5 | 36 | 24 | 18 |
| $F+$ (N) | 147.95 | 131.79 | 141.52 | 132.94 | 139.18 | 149.14 | 121.55 | 153.74 | 145.97 |
| $F-$ (N) | 185.52 | 162.44 | 180.59 | 177.09 | 174.01 | 177.46 | 139.98 | 189.38 | 199.19 |

### 3.2. Wear and Life Test Results

The relevant parameters for wear and lifespan testing are provided in Table 4, where $Y$ is the test number, $m$ is the amount of wear, and $L$ is the life of the piston. The results of the wear and life tests are shown in Table 4. The piston with cylindrical pit arrays could significantly reduce the wear rate $w$ and improve the service life $L$ of the piston. Compared with piston S-1, the $w$ value of pistons 4 and 13 reduced by more than double, and $v$ increased by more than 30%. Compared with piston S-2, the $w$ value of piston 22 experienced a substantial decrease, reducing by approximately 50%. Meanwhile, $v$ demonstrated a significant enhancement, with an increase of at least 48%. Since $H$ for piston 25 was relatively large and had a certain influence on the wear performance and service life of the piston, the changes in $w$ and $v$ were not significant. In conclusion, all the textured pistons demonstrated a decrease in $w$ with varying degrees, while $v$ exhibited an increase to different extents. Notably, the wear amount $m$ for pistons 4, 13 and 22 reduced by over 50%, resulting in an improvement in their lifespan $L$ of at least 30% or more.

**Table 4.** Results of wear and life tests.

| $Y$ | $m_0$ (g) | $m_f$ (g) | $m$ (g) | $w$ | $L$ (min) | $v$ |
|---|---|---|---|---|---|---|
| S-1 | 140.96 | 134.42 | 6.54 | 4.64% | 1139 |  |
| 4 | 140.65 | 137.86 | 2.79 | 1.98% | 1648 | 44.69% |
| 13 | 139.17 | 136.60 | 2.57 | 1.84% | 1519 | 33.36% |
| S-2 | 144.82 | 139.51 | 5.31 | 3.66% | 1161 |  |
| 22 | 144.56 | 141.98 | 2.58 | 1.78% | 1721 | 48.23% |
| 25 | 143.62 | 138.72 | 4.90 | 3.41% | 1333 | 14.81% |

### 3.3. Thermal Imaging Test Results

According to the average temperature at different times, the time-temperature curve was plotted as shown in Figure 13. It can be seen that within 100 min, the temperature increased rapidly from below 30 °C to above 45 °C. From 100 min to 300 min, the temperature rose slowly. Being affected by complicated working conditions, the temperature would

fluctuate with time and be in a relatively stable state after 300 min. During the continuous and stable operation of the mud pump (300–600 min), compared with the standard piston, the temperature of the friction pair of the piston with cylindrical pit arrays was significantly reduced. The temperature curves indicate that the curves of the textured pistons were consistently below those of the corresponding standard pistons. The temperature of piston S-1 was obviously higher than that of pistons 4 and 13. Similarly, the temperature of piston S-2 was significantly higher than those of pistons 22 and 25. In comparison with the corresponding standard pistons, the temperature of nearly all textured pistons was reduced by around 5–10 °C. The higher the temperature, the more severe the friction and wear of the piston, and the shorter its service life will be.

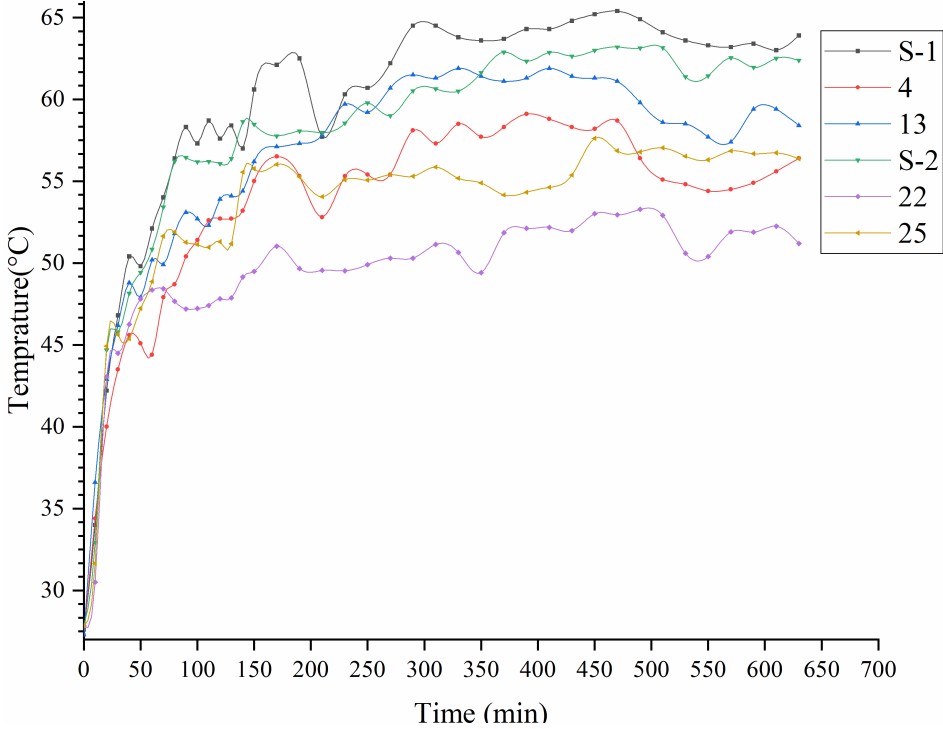

**Figure 13.** Time–temperature curve.

## 4. Friction and Wear Mechanism

Figure 14 shows the deformation of the standard pistons and textured pistons 1–27. It can be seen that the maximum deformation of the piston was concentrated at the edge of the rubber part. The distribution of deformation was in a ring shape, decreasing gradually from top to bottom. The wear of the piston was mainly concentrated in the position with large deformation, which is consistent with the results of the wear test. Compared with the standard pistons, the maximum deformation of the textured pistons was reduced to some extent. The primary cause for this occurrence is that the presence of cylindrical pit arrays can provide more compression space on the piston surface, and each cylindrical pit structure can withstand a higher degree of deformation, alleviating the phenomenon of concentrated deformation distribution at the piston's lip. We performed an in-depth analysis of textured pistons 4, 13, 22 and 25, with their maximum deformation data ($x$) presented in Table 5. It becomes apparent that $x$ for these four pistons ascended with an increase in $X$. Specifically, piston 4 witnessed the highest reduction in its maximum deformation, tallying a decrease of 25.49%. Meanwhile, the other three pistons showed less pronounced reductions, with all falling under 10%, specifically at 7.25%, 2.51% and 1.41%.

**Table 5.** Maximum deformation and maximum equivalent stress.

| Y | S | 4 | 13 | 22 | 25 |
|---|---|---|----|----|----|
| *x* (mm) | 1.1701 | 0.8607 | 1.0713 | 1.1261 | 1.1388 |
| *y* (MPa) | 0.85131 | 0.71788 | 0.92184 | 0.88829 | 0.93348 |

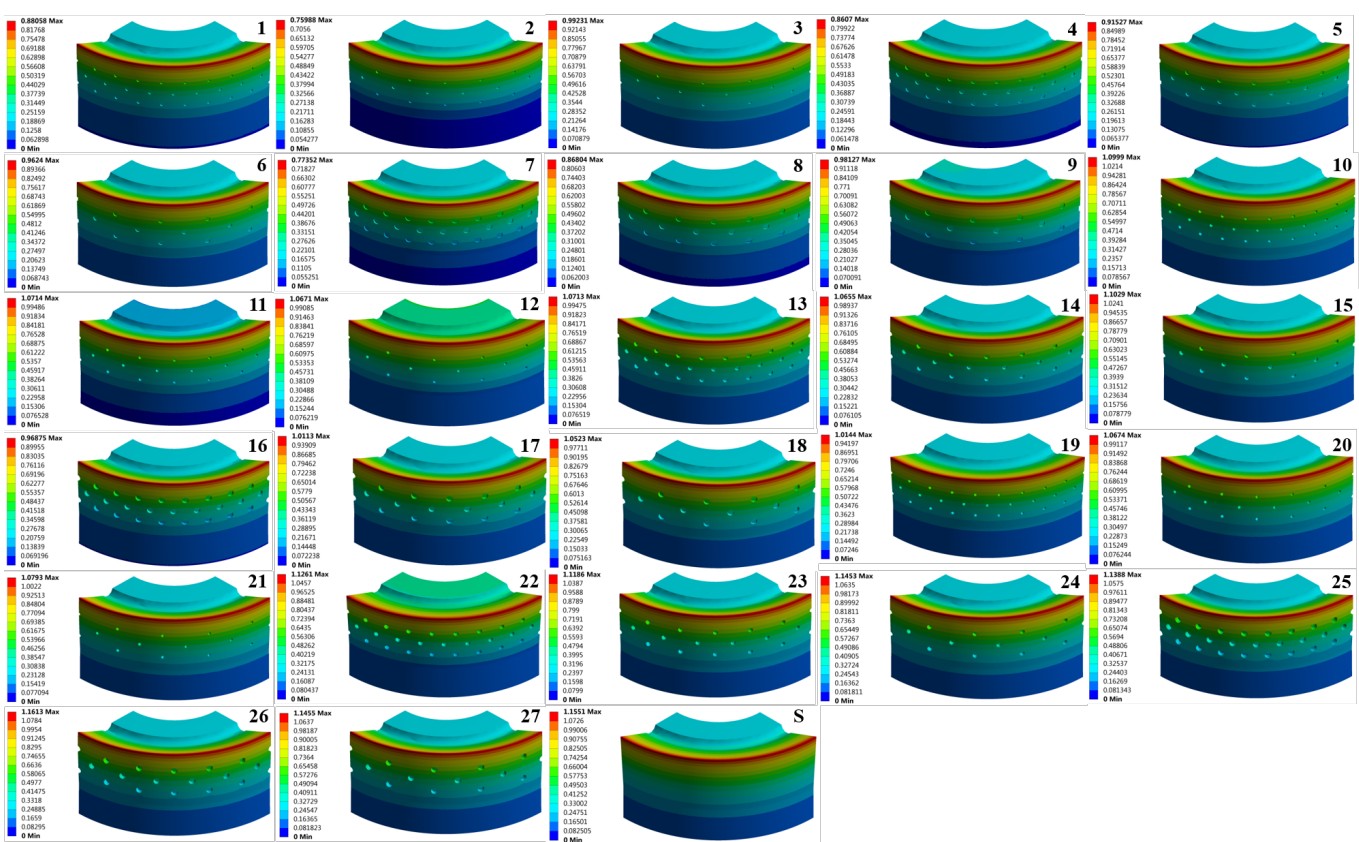

**Figure 14.** Deformation of standard and textured pistons.

Figure 15 shows the equivalent stress of the standard pistons and textured pistons 1–27. The maximum equivalent stress of the standard piston appears on the upper surface. Excessive equivalent stress can cause serious extrusion and premature failure of the piston. The maximum equivalent stress of the piston with cylindrical pit arrays is distributed near the cylindrical pit, which can reduce the phenomenon of piston squeezing. The maximum equivalent stress of textured pistons 1–9 was smaller than those of the standard pistons. The maximum equivalent stress of textured pistons 19–27 increased to a certain extent compared with the standard pistons, while the equivalent stress of textured pistons 10–18 increased in some cases and decreased in others. This may be related to the depth of the cylindrical pit texture. Therefore, in the analysis of the equivalent stress of the textured piston, we concluded that *H* for the textured pistons had the most obvious impact on the equivalent stress. *H* in this article should be less than 0.5 mm to reduce the equivalent stress of the textured piston and reduce the impact of the cylindrical pit texture array on the surface strength of the piston. We performed an in-depth analysis of textured pistons 4, 13, 22 and 25, with their maximum equivalent stress (*y*) presented in Table 5. Compared with the standard piston, *y* for piston 4 was significantly reduced, while *y* for pistons 13, 22 and 25 did not change much. The *y* value of textured piston 4 decreased by 15.67%, while *y* for the other three textured pistons increased by no more than 10%. Since the equivalent stress distribution of the textured piston will be dispersed to each cylindrical pit texture, the depth and size of the cylindrical pit array will significantly affect the equivalent stress of the piston. Too deep and too large of a cylindrical pit size will destroy the strength of the

piston's surface to a certain extent, thus affecting the service life of the piston. Therefore, our optimized piston needs to be verified in a life test.

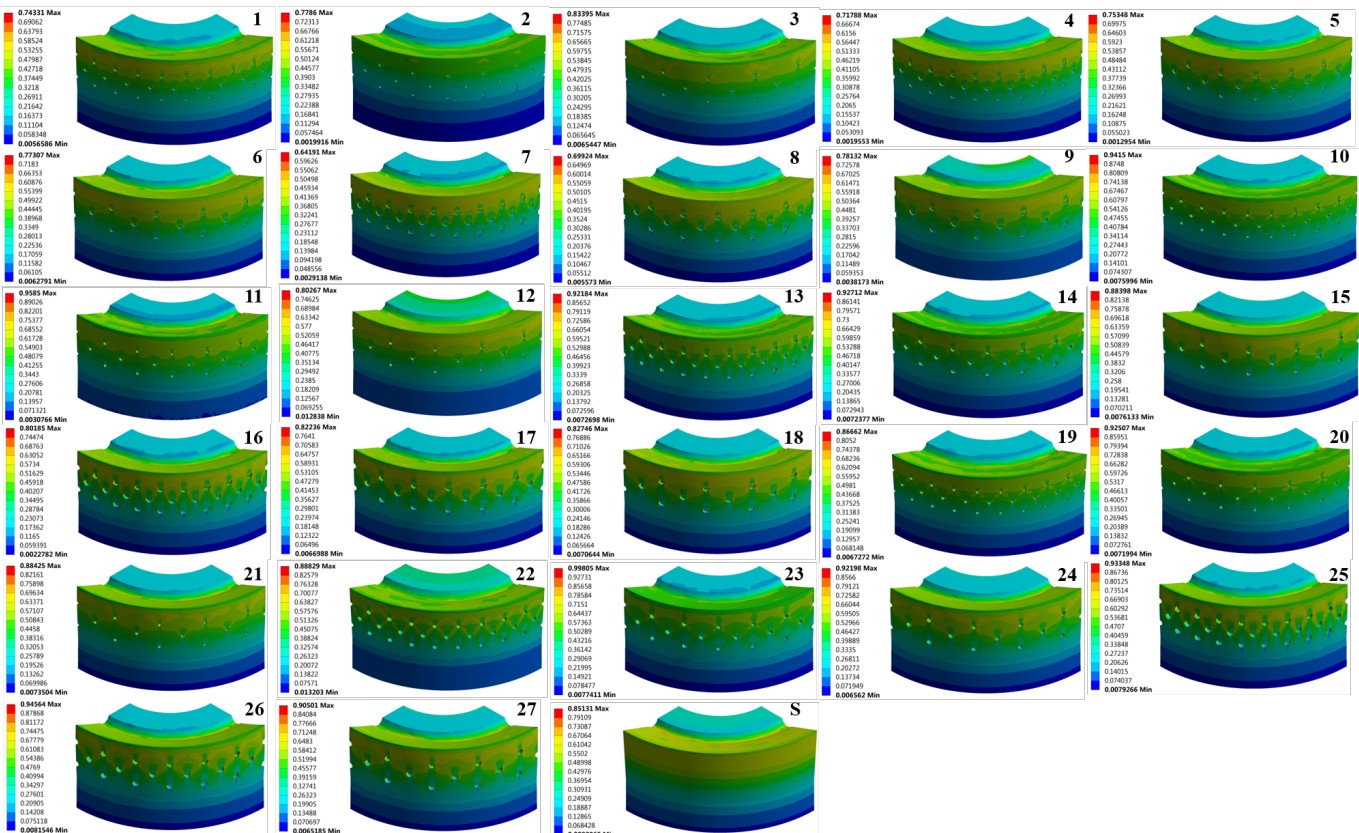

**Figure 15.** Equivalent stress of standard and textured pistons.

As shown in Figure 16, the contact pressure contour maps of the 27 textured pistons and the standard piston are presented. Similar to the deformation, the contact pressure distribution of the textured pistons also exhibited a ring-shaped pattern with a gradual decrease from top to bottom. In comparison with the standard piston, the maximum contact pressures of the textured pistons showed slight variations, with some increasing and others decreasing. However, these changes are not significant, and the contact pressure at the lip area of the textured pistons generally met the sealing requirements of the mud pump. Among them, our focus primarily lies on textured pistons 4, 13, 22 and 25. Compared with the standard piston, these four pistons displayed varying changes in their maximum contact pressures. Notably, the contact pressure of textured piston 4 experienced a substantial decrease of 16.81%. The maximum contact pressures of the other three textured pistons showed negligible variations, with percentage increases or decreases of around 5%. These fluctuations in contact pressure may have certain implications for piston sealing. However, if the pistons do not exhibit any leakage during lifespan testing, then this indicates that the contact pressure is completely satisfactory.

Abrasive particles in the gap between the mud pump piston and the cylinder liner will cause high contact stress, increase the friction and wear of the friction pair and reduce the service life of the piston and cylinder liner. Cylindrical pit arrays can not only capture part of the abrasive particles but also provide secondary lubrication for the friction pair. Figure 17 shows the piston and cylinder liner after testing. It can be seen that the rubber edges of the standard piston were severely worn and all fell off, and the surface of the cylinder liner had many obvious scratches. However, the rubber edge part of textured piston 4 did not completely come off, and the surface of the cylinder liner had fewer

scratches. Part of the abrasive particles is stored in cylindrical pits, which can reduce the three-body wear of the friction pair.

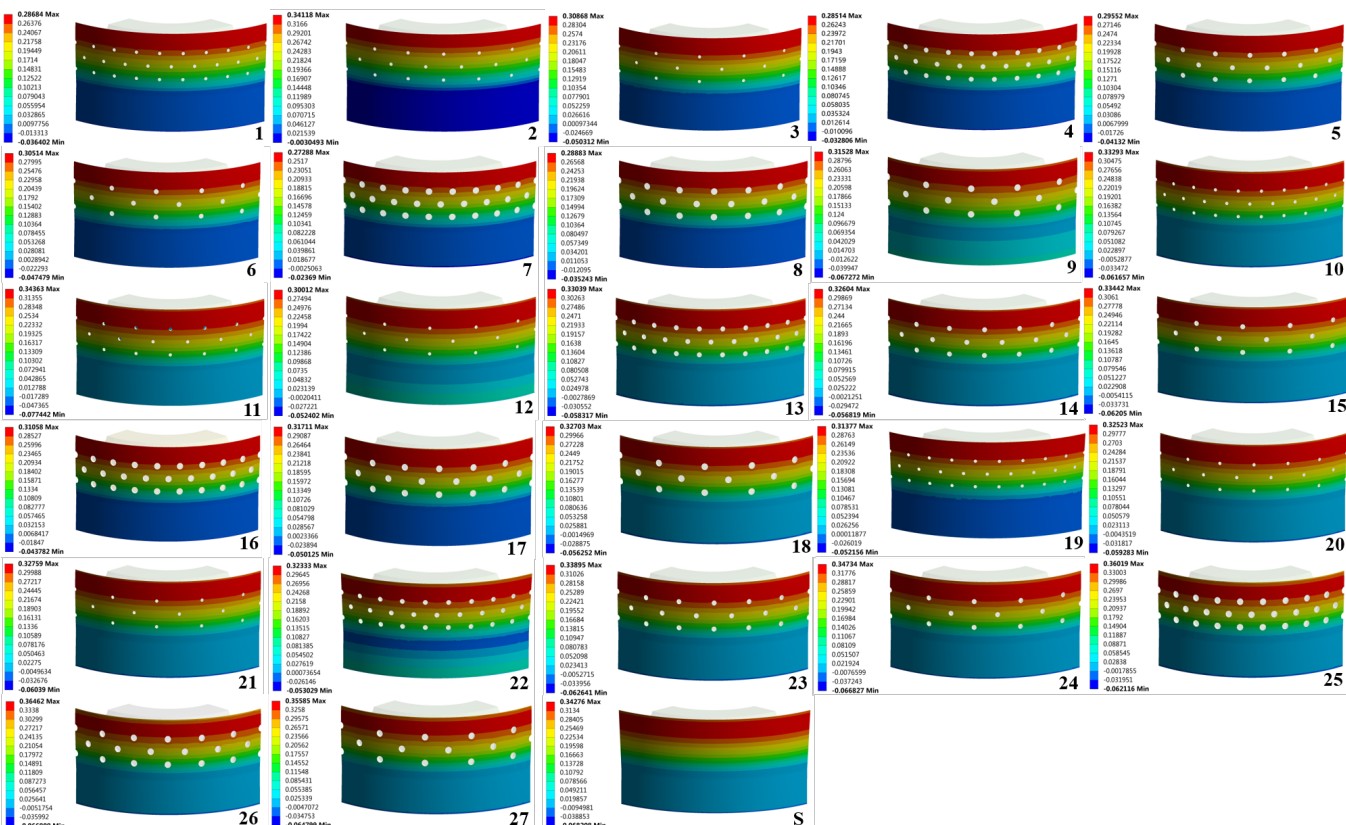

**Figure 16.** Contact pressure of standard and textured pistons.

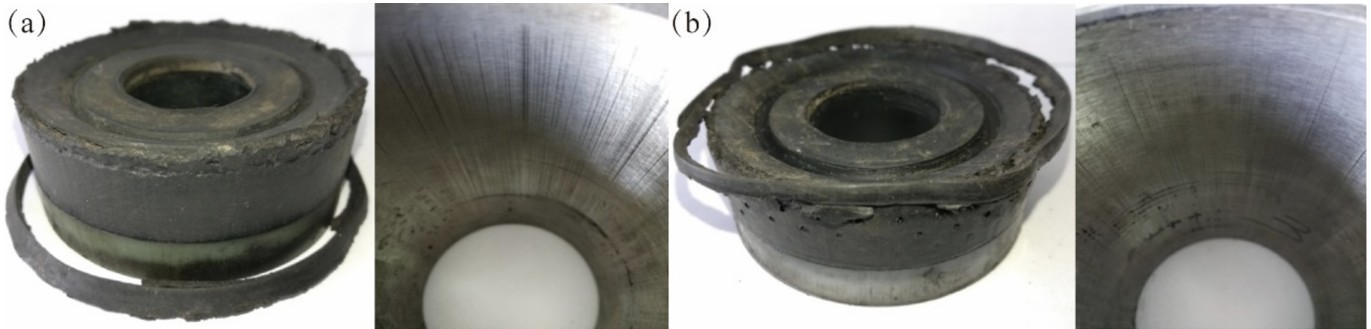

**Figure 17.** Piston and cylinder liner after test: (**a**) standard piston and (**b**) piston 4.

## 5. Conclusions

(1) To improve the friction performance of the piston, the diameter ($D$) of the cylindrical pit should not be too large or too small. With the increase in $D$, both $F+$ and $F-$ showed an initial upward trend followed by a decrease, reaching the minimum at $D$ = 1.5 mm. The best friction performance of the piston was achieved with $H$ = 0.5 mm. An excessively large cylindrical pit depth ($H$) adversely affects the friction performance. When the number ($N$) of cylindrical pits set to 36, the friction performance of the piston was better.

(2) Under the conditions of a pressure of 0.5 MPa and 200 strokes per minute, the piston with cylindrical pit arrays can significantly reduce the wear rate ($w$), increase the service life ($L$) of the piston and also obviously reduce the temperature of the friction pair compared with the standard piston. The wear amount $m$ of pistons 4, 13 and 22 was

reduced by over 50%, resulting in an improvement in their lifespan $L$ of at least 30% or more. The temperature of the textured pistons was reduced by around 5–10 °C.

(3) The maximum deformation ($x$) of the piston with cylindrical pit arrays was small. Piston 4 showed the most significant decrease in its maximum deformation, with a reduction of 25.49%. And the maximum equivalent stress ($y$) was distributed near the cylindrical pit, which could reduce the phenomenon of piston squeezing. The maximum equivalent stress of textured piston 4 decreased by 15.67%, while the $y$ values of the other three textured pistons increased by no more than 10%. The maximum contact pressures of the textured pistons showed slight variations, with some increasing and others decreasing.

(4) The cylindrical pit arrays can not only provide secondary lubrication but also capture part of the abrasive particles to reduce the three-body wear of the friction pair.

**Author Contributions:** Methodology, T.G.; Validation, T.G.; Resources, Y.W.; Data curation, H.C.; Writing—original draft, T.G.; Writing—review & editing, H.C. and D.T. All authors have read and agreed to the published version of the manuscript.

**Funding:** This research was supported by the National Natural Science Foundation of China (Grant No. 52205309) and the Nanjing University of Science and Technology under Award No. AE89991/383.

**Data Availability Statement:** The data that support the findings of this study are available from the corresponding author upon reasonable request.

**Acknowledgments:** T. Gao would like to acknowledge the support from the Project of Philosophy and Social Science Research in Colleges and Universities in Jiangsu Province under Award No.2023SJYB0015 and the 2022 High Level Innovation and Entrepreneurial Research Team Program in Jiangsu (JSSCBS20220262). Y. Wang would like to acknowledge the support from the Fundamental Research Funds for the Central Universities (No. 30922010719). We would like to thank Gao, Chen and Tang for their valuable contributions to this research.

**Conflicts of Interest:** The authors declared no potential conflict of interest with respect to the research, authorship or publication of this article.

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
