# Peer review of "Inspired by Earthworms and Leeches: The Effects of Cylindrical Pit Arrays on the Performance of Piston-Cylinder Liner Friction Pairs"

_applsci, doi:10.3390/app132011580_

Round 1
Reviewer 1 Report
The article focuses on investigation of the friction and wear performance of the piston-cylinder liner friction pair like in mud pumps.
The authors have thoroughly investigated the question and provided valuable insights into the distribution of vapor volume fraction obtained and the influence of mass flux, heat flux, saturation temperature, and vapor quality on heat transfer coefficient (HTC).
The methodology employed in this study was sound and appropriate for addressing the research question (finite element analysis + experiment). The “results and discussion” section provides a clear interpretation of the results and their relevance to the research question. The writing style is clear and concise, making the article accessible to a broad audience.
Overall, this manuscript makes a significant contribution to the corresponding area and is worthy of publication but there are some points in the article that need to be clarified and refined:
1. Improve the abstract: "The results show that the pistons with cylindrical pit arrays have excellent friction and wear performance, less heat generation by friction (How much less? Please provide concrete results), longer life (How much longer is the life?), and less scratches on the cylinder liner."
2. To increase the relevance of the article, the authors can add modern articles in the introduction that study the problems of the piston-cylinder pair. For example:
- wear and jamming of spool pairs in hydraulic drives
https://doi.org/10.3103/S1068366620060021
- experimental investigation on effects of positive texturing on friction and wear reduction of piston ring/cylinder liner system
https://doi.org/10.1016/j.matpr.2020.04.424
- tribological and mechanical performance of coatings on piston to avoid failure—a review
https://doi.org/10.1007/s11668-022-01436-3
3. The conclusions contain general wording. Please concretize the results to justify your research:
1) To improve the friction performance of the piston, the diameter (D) of the cylindrical pit should not be too large or too small (which size?). An excessively large cylindrical pit depth (H) adversely affects the friction performance (How does it affect and what is the limit value of large cylindrical pit depth?). When the number (N) of cylindrical pits is large, the friction performance of the piston is better (what is the limit number?).
2) Compared with the standard piston, the piston with cylindrical pit arrays can significantly reduce the wear rate (w) (How much to reduce and under what conditions?) and increase the service life (L) of the piston (How much to increase and under what conditions?), and can also obviously reduce the temperature of the friction pair (How much to reduce and under what conditions?).
3) The maximum deformation of the piston with cylindrical pit arrays is small (What is the numerical equivalent?), and the maximum equivalent stress is distributed near the cylindrical pit, which can reduce the phenomenon of piston squeezing.
4) The cylindrical pit arrays can not only provide secondary lubrication, but also capture part of the abrasive particles to reduce the three-body wear of the friction pair.
Reviewer 2 Report
The paper is very interesting and well thought out. It is necessary to correct the paper according to the proposals and suggestions of the reviewer.
· At the end of the introduction, please state the main contribution of the paper and highlight how this paper differs from similar studies in this field. Explain why this paper should be published.
• It would be good to compare the results of the research with the results of other authors.
• References must be expanded.
· In the introduction part add more about planetary gear types, etc, see: M. Stanojević, R. Tomović, L. Ivanović, B. Stojanović, Critical Analysis of Design of Ravigneaux Planetary Gear Trains. Applied Engineering Letters, 7(1), 2022: 32-44.https://doi.org/10.18485/aeletters.2022.7.1.5. Also, there are authors that investigate optimization of planetary gears, see: S. Miladinović, S. Veličković, M. Novaković, APPLICATION OF TAGUCHI METHOD FOR THE SELECTION OF OPTIMAL PARAMETERS OF PLANETARY DRIVING GEAR, Applied Engineering Letters, 1(4), 2016: 98-104.
· What does MP3 Hybrid, P3 hybrid and PCX eHEV mean?
· Where abbreviation first appears, there should be as well full name in the brackets or vice versa.
· What is the type of planetary gear set observed in the paper?
· Electric machine is electromotor? Why do you call it electric machine?
· Was the efficiency of demonstrator vehicle measured?
· What type of vehicle is demonstrator vehicle? Is it two-wheel, three or four-wheel? Based on figure 5 it looks like 3- or 4-wheel vehicle.
· It would be good to compare the results of the research with the results of other authors. Add more discussion.
· Expand the concluding remarks.
Minor editing of English language required.
Reviewer 3 Report
This is a very poorly written paper. A major revision.
The submitted manuscript must be extensively revised. As general comments that need to be corrected are:
· The writing technique must be adapted to the scientific style of writing;
· It is necessary to exclude the text taken from the prospectus or similar documents;
· The text has no scientific traceability;
· The material and work method chapter must be better presented. It should be said which scientific methods were used in the research, software, techniques, tools and test tables, etc.;
· The abstract must be better presented and described. The writing style should be adapted to the academic style. Avoid the words "Then and Finally"
· Lines 33-34: When citing an author in a sentence, it should be cited at the end of the sentence: For instance, Zhang et al..............
· Lines 36-37: When the author is mentioned in the sentence, it should be quoted at the end of the sentence: Moreover, Zhou and his team..........
· Lines 33-34: Zhou and his team cannot be cited .......... it should be Zhou et al.,
· Lines 44-46: The sentence is not quoted, and the authors are listed: Moreover, Jia et al.
· Line 59: Incorrect citation of "Etsion et al. found". There is only one author of the paper.
· Line 69: At the beginning of the sentence You should not say knowledge, but research.
· Chapter 2. Experimental should read Material and Methods. Chapter 2 is not well presented. This chapter must be scientifically presented better. It should be said which scientific methods were used in the research, software, techniques, tools and test tables, etc.;
· Line 96: Scientific works should not be written in Subject Pronouns "we utilize"...etc
· Lines 98-99: Unclear sentences
· Lines 112-113: The sentence is inappropriate for a scientific paper. This sentence is taken from a prospect or similar document that describes the operation of the device.
· Figures 2 and 3 should be presented in those chapters where they are first mentioned in the paper.
· Line 128: Why "mainly". What are the other alternatives?
· Why are Figures 6 and 7 shown in the Research Results chapter?
· Lines 159 and 161: Extract the formulas from the text and number them. Why are these formulas not given in the Materials and Methods chapter?
· Lines 171-173: The description of the work methodology should not be in the Search Results Chapter.
· Lines 174-175: An inappropriate sentence for a scientific paper. Again, material from the prospect or similar is used.
· Line 172: Correct: We use Fluke...
· After Chapter 3 Results and discussion, Chapter 4. Friction and wear mechanism cannot be given
· Lines 189-192: The text has no place in this part, it should be in the introduction.
· Most of the text from Chapter 4 must be found in the Material and Method of Work chapter.
· Line 262: Correct: We conducted...
· The conclusion lacks an introductory sentence, e.g. The key results of the research were:
· Expand references with the following research:
Prevention of Water Ingress in Hydraulic Systems. Applied Engineering Letters, 6(3), 2021: 99–104,
Milojević, Saša, et al. "Correlation between Emission and Combustion Characteristics with the Compression Ratio and Fuel Injection Timing in Tribologically Optimized Diesel Engine." Tehnički vjesnik 29.4 (2022): 1210-1219.
Round 2
Reviewer 3 Report
I suggest work for the press. the authors accepted all proposals and suggestions